Identification of candidate long non-coding RNAs and mRNAs associated with heart aging in mice

Kuai Zheng 1
Li Zheng 2
Jia Jianguo 3
Chen Yongle 4
Zhang Xiaoyi 1
Zhang Jianhui 3
Ye Yangli 1
Gao Lihong 1
Li Ling 1
Hu Yu hu.yu@zs-hospital.sh.cn 1 5
1 Department of Geriatrics, Zhongshan Hospital, Fudan University , Shanghai , China
2 Clinical Science Institute, Zhongshan Hospital, Fudan University , Shanghai , China
3 Department of Cardiology, Shanghai Institute of Cardiovascular Disease, Zhongshan Hospital, Fudan University , Shanghai , China
4 Department of Echocardiography, Shanghai Institute of Cardiovascular Diseases, Shanghai Institute of Medical Imaging, Zhongshan Hospital, Fudan University , Shanghai , China
5 Center for Evidence Based Medicine and Clinical Epidemiology, Zhongshan Hospital, Fudan University , Shanghai , China
Lasseigne Brittany
Electronic publication date: 2025 Dec 2
Publication date: 2025
Volume: 13
Electronic Location ID: e20433
Received 2025 May 30; Accepted 2025 Oct 30
Copyright: ©2025 Kuai et al.
Copyright year: 2025
Copyright holder: Kuai et al.
License: This is an open access article distributed under the terms of the Creative Commons Attribution License, which permits unrestricted use, distribution, reproduction and adaptation in any medium and for any purpose provided that it is properly attributed. For attribution, the original author(s), title, publication source (PeerJ) and either DOI or URL of the article must be cited.
License URL: https://creativecommons.org/licenses/by/4.0/

Keywords: Naturally aging mice, Cardiac aging phenotype, Microarray, Non-coding RNAs

Funding: The authors received no funding for this work.

==============================
Background

Understanding the molecular mechanisms underlying cardiac aging may uncover novel therapeutic targets for age-related cardiovascular disease. Long non-coding RNAs (lncRNA), which regulate cell differentiation and disease progression, are emerging as promising diagnostic biomarkers and therapeutic candidates. However, their expression profiles and functional roles in the aging heart remain poorly characterized.

Methods

Male C57BL/6 wild type mice aged 20 months (aged group) and 3 months (young group) underwent transthoracic echocardiography to evaluate cardiac function. Myocardial aging phenotypes were assessed using hematoxylin-eosin, Masson’s trichrome, terminal deoxynucleotide transferase dUTP nick end labeling (TUNEL), and senescence-associated β-galactosidase staining. Transcriptomic profiling was performed using a lncRNA-focused microarray platform to identify differentially expressed lncRNAs and mRNAs in heart tissues.

Results

Aged mice showed increased heart weight/body weight and heart weight/tibia length ratios. Both interventricular septum in systole and left ventricular posterior wall in diastole were elevated, while ejection fraction and fractional shortening remained unchanged. The Tei index was significantly higher, suggesting impaired myocardial performance. Histological staining revealed enlarged cardiomyocytes, increased myocardial fibrosis, enhanced apoptosis, and greater senescence-associated β-galactosidase activity. Microarray analysis identified distinct age-related expression patterns of lncRNAs and mRNAs in the heart.

Conclusions

Cardiac aging is characterized by structural and functional remodeling, accompanied by transcriptional reprogramming involving both lncRNAs and mRNAs. These changes offer insights into potential molecular mechanisms and provide candidate regulatory targets for diagnosis and intervention in age-related heart disease.

Introduction

With the rapid growth of the global elderly population, cardiac aging has emerged as one of the greatest healthcare challenges worldwide. Cardiac aging is characterized by progressive structural and functional changes that predispose individuals to cardiovascular diseases, the leading cause of death among older adults. Notably, intrinsic cardiac aging occurs even without systemic cardiovascular risk factors such as smoking, dyslipidemia, hypertension, or diabetes, underscoring the importance of understanding its molecular basis.

Traditionally, aging has been viewed as an inescapable, time-dependent decline in physiological function—an irreversible process tied to chronological age (Aviv, 2002). However, growing evidence reveals substantial variation in the pace of aging across species, individuals, and even organs within the same organism (Campisi et al., 2019). This emerging distinction between biological and chronological aging suggests that aging may be modifiable and potentially subject to intervention.

Epigenetic dysregulation has been recognized as a central hallmark of biological aging (López-Otín et al., 2023). Among the epigenetic modulators, non-coding RNAs (ncRNAs), particularly long non-coding RNAs (lncRNAs), play critical roles in regulating gene expression and maintaining transcriptomic stability. Recent studies have highlighted the regulatory roles of ncRNAs, including microRNAs (miRNAs) and lncRNAs, in cardiovascular aging (Poller et al., 2018; Boon et al., 2016). For instance, miR-34a, miR-29, and miR-199 promote cardiac aging through mechanisms involving apoptosis, fibrosis, and impaired tissue regeneration (Boon et al., 2013; Lyu et al., 2018; Eulalio et al., 2012). Similarly, lncRNAs such as H19 and MALAT1 have been implicated in cardiomyocyte senescence, myocardial remodeling, and functional decline via transcriptional regulation, chromatin modulation, and miRNA interaction (Zhang et al., 2019; Zhu et al., 2019).

Animal models, especially mice, are widely used in aging research due to their genetic manipulability, short lifespan, and well-conserved cardiac structure and function relative to humans (Hasty & Vijg, 2004). Extensive studies in aged mouse models have revealed conserved phenotypes of cardiac aging, including increased myocardial stiffness, impaired diastolic relaxation, cardiomyocyte hypertrophy, interstitial fibrosis, mitochondrial dysfunction, and transcriptional reprogramming (Chiao & Rabinovitch, 2015). However, previous transcriptomic studies have focused on isolated protein-coding genes or selected ncRNAs. Consequently, the comprehensive transcriptomic landscape and the potential regulatory interactions between lncRNAs and mRNAs in cardiac aging remain poorly defined.

Given these gaps, the present study aimed to systematically characterize both phenotypic and transcriptomic changes in naturally aging mouse hearts. Specifically, we evaluated cardiac function via echocardiography, conducted histological analyses, and employed microarray to simultaneously identify differentially expressed lncRNAs and mRNAs. This integrated approach provides novel insight into age-associated cardiac remodeling at both functional and molecular levels.

Although lncRNAs and mRNAs were derived from the same microarray dataset, they were analyzed separately due to fundamental differences in their biological roles, regulatory mechanisms, and annotation processes. Whereas mRNAs encode proteins directly mediating cellular functions, lncRNAs primarily regulate gene expression through mechanisms involving chromatin remodeling, transcriptional interference, and post-transcriptional modulation (Zhao et al., 2025; Abbas & Gaye, 2025). Conducting separate analyses enabled a more targeted interpretation of expression patterns and facilitated the preliminary construction of potential regulatory relationships between lncRNAs and mRNAs implicated in cardiac aging.

Methods

Animal model and grouping

C57BL/6 wild-type male mice were used in this study. Young mice aged 10 weeks were obtained from Shanghai Jiesijie Laboratory Animal Co., while elderly mice (approximately 18 months old) were sourced from Jiangsu Lingfei Biotechnology Co. The latter were raised to 20 months of age at our facility to establish a natural aging model. Animals were grouped based on age into:

• 3m group:3-month-old male mice (n = 5)

• 20m group:20-month-old male mice (n = 5).

This sample size has been widely used in similar exploratory studies comparing age-related differences in cardiac molecular expression and is considered sufficient to identify significant differential expression of target genes, while adhering to ethical guidelines minimizing animal usage (Keele et al., 2023). Although no formal power calculation was performed prior to the study due to its exploratory nature, the chosen number aligns with standard practices in molecular and transcriptomic profiling studies, allowing adequate statistical power for detecting biologically meaningful differences between age groups.

Mice were housed individually in specific pathogen-free (SPF) conditions at the Laboratory Animal Center of Zhongshan Hospital, Fudan University. The environment was maintained at 23 ± 1 °C with 50% ± 10% humidity and a 12-hour light/dark cycle. Mice had ad libitum access to sterile water and standard rodent chow (Shanghai SLAC Laboratory Animal Co., Ltd.) with environmental enrichment provided. The study was approved by the Animal Ethics Committee of Zhongshan Hospital, Fudan University (Approval No. 2021-20). Single housing was used for all animals.

Mice were included if they appeared healthy, with no signs of illness or weight loss. No animals or data points were excluded. Due to the distinct physical characteristics of age groups, full randomization and blinding were not feasible for animal handling, but all outcome assessors for molecular analyses were blinded to group identity.

Echocardiographic assessments were conducted 24–48 h prior to euthanasia to allow sufficient recovery from isoflurane anesthesia while minimizing physiological variability. All mice were euthanized by cervical dislocation without prior anesthesia to avoid anesthetic-induced transcriptomic alterations. Cardiac tissue was immediately harvested and snap-frozen in liquid nitrogen for microarray and downstream analysis.

Echocardiography

Mice were anesthetized with 1% isoflurane (R510-22, RWD Life Science) and examined using a Vevo 2100 High-Resolution Imaging System (FUJIFILM Visual Sonics, Toronto, Canada) equipped with an 707B 30 MHz probe (Moreno-Fernandez et al., 2021). Parasternal long- and short-axis views were acquired. Parameters including heart rate (HR), interventricular septum in systole and diastole (IVSs, IVSd), left ventricular internal dimension in systole and diastole (LVIDs, LVIDd), left ventricular posterior wall in systole and diastole (LVPWs, LVPWd), rejection fraction (LVEF%) and fractional shortening (FS%) were measured using Vevo LAB software (Version 3.2.6).

Cardiac diastolic function was assessed by the myocardial performance index (MPI or Tei index) (Lakoumentas et al., 2005). Calculated as of Tei index = (isovolumic relaxation time (IVRT) + isovolumetric contraction time (IVCT))/ejection time (ET).

Histological staining

Tissue sampling for histological analyses occurred immediately following euthanasia, at a single time point for each animal.

For hematoxylin-eosin (HE) staining, cardiac tissues were fixed with 10% paraformaldehyde (PFA; Sigma-Aldrich, Cat# P6148) and embedded in paraffin. A 4 µm was stained with hematoxylin for 2 min, washed, and then counterstained with eosin. For Masson staining, tissues were sequentially treated with Masson trichrome kit reagents (Solarbio, Cat# G1340), including phosphotungstic acid, bright green, and acetic acid washes. Slides were dehydrated, cleared, mounted, and visualized under a light microscope.

Apoptosis detection

Apoptosis was performed using a terminal deoxynucleotide transferase dUTP nick end labeling (TUNEL) assay kit (Roche, Cat# 11684795910). Cardiac sections were permeabilized, incubated with TUNEL reaction mixture, and developed with 3,3’-Diaminobenzidine (DAB) substrate. Positive apoptotic nuclei were visualized using light microscopy.

Senescence-associated β-galactosidase staining

Tissues were fixed with 10% PFA and embedded in OCT compound. Frozen sections (10 µm) were cut, air-dried, and treated for β-galactosidase staining. After antigen retrieval and blocking, sections were incubated with anti-β-galactosidase (Abcam, cat#ab9361, 1:500) overnight at 4 °C, followed analyzed by Image J software (v1.53k,National Institutes of Health, Bethesda, MD, USA).

Microarray and transcriptome analysis

Myocardial tissue samples were collected from 3-month-old and 20-month-old mice. Each animal was sampled at a single time point. Total RNA was extracted using TRIzol reagent (Invitrogen, USA), and its quantity and purity was assessed using NanoDrop One spectrophotometry (Thermo Fisher Scientific, USA). RNA integrity was confirmed by denaturing agarose gel electrophoresis.

Gene expression profiling was performed using the Agilent-085983 Arraystar Mouse LncRNA Microarray V4.0 platform (GPL26962), which enables simultaneous detection of both long non-coding RNAs (lncRNAs) and protein-coding mRNAs. Sample labeling, hybridization, washing, and scanning were conducted by Aksomics Bio (Shanghai, China), following the manufacturer’s standard protocol.

Briefly, total RNA was treated to remove ribosomal RNA and amplified into fluorescently labeled complementary RNA (cRNA) using the Arraystar Flash RNA Labeling Kit. The labeled cRNAs were hybridized to the microarray slides and scanned with the Agilent G2505C Scanner. Raw signal intensities were extracted using Agilent Feature Extraction Software (version 10.7.3.1).

Downstream data processing and statistical analyses were performed using R version 3.6.0 and Bioconductor release 3.10. Raw data were background-corrected and quantile-normalized. Probes flagged as “Present” or “Marginal” in at least 50% of samples were retained for further analysis. Differentially expressed genes (DEGs), including both lncRNAs and mRNAs, were identified using the limma package (version 3.42.2), applying linear models and empirical Bayes moderation. Genes with an adjusted p-value (false discovery rate, FDR) <0.05 and —log2 fold change— ≥ 1 were considered statistically significant.

Functional enrichment analyses, including Gene Ontology (GO) terms and Kyoto Encyclopedia of Genes and Genomes (KEGG) pathways, were conducted using the clusterProfiler package (version 3.14.3). All annotated genes from the platform were used as the background gene set.

No technical replicates were included. Five biological replicates per group were used to ensure statistical robustness and account for biological variability.

The microarray data generated in this study have been deposited in the NCBI Gene Expression Omnibus (GEO) under accession number GSE256034.

Protein–protein interaction network and functional annotation

To investigate potential functional relationships and molecular interactions among the most DEGs mRNA associated with cardiac aging, we selected the top 100 DEGs based on adjusted p-value and log2 fold-change. Protein–protein interaction (PPI) network analysis was performed using the STRING database (version 12.0; https://string-db.org) with the organism set to Mus musculus and a medium confidence score threshold of 0.4.

The network statistics, including average node degree and enrichment p-values, were extracted directly from the STRING output. Unconnected nodes and sparse interactions were retained to reflect the full distribution of DEGs.

In addition, we selected the top 20 DEGs for individual functional annotation. Gene symbols were queried using the GeneCards, UniProt, and PubMed databases to obtain descriptions of biological roles, especially in pathways related to immune response, mitochondrial function, cellular transport, or transcriptional regulation.

Statistics

As an exploratory observational study, formal sample size calculation based on a primary outcome measure was not performed. The chosen sample size was determined based on previously published literature and preliminary experimental experience, ensuring sufficient sensitivity for identifying biologically meaningful differences in molecular profiles and phenotypic characteristics between age groups.

Data are expressed as mean ± standard error or standard deviation. For comparisons of continuous variables between groups, the student’s t-test was used for Gaussian data and the Mann–Whitney test for non-Gaussian data.

For microarray transcriptomic analyses, differentially expressed genes were identified using appropriate statistical tests with multiple hypothesis testing correction. Adjusted p-values were calculated using the Benjamini–Hochberg false discovery rate (FDR) method, and genes with FDR <0.05 and —log2 fold change— ≥ 1 were considered statistically significant.

Statistical analyses were performed using SPSS 26.0 (SPSS Inc., Chicago, Illinois, U.S.A.) and GraphPad Prism 9.20 software (GraphPad Software, Inc., La Jolla, CA, U.S.A.) and transcriptomic analyses were conducted using R (version 3.6.0) with standard bioinformatics packages. P-values <0.05 were considered significant.

Results

Cardiac phenotype in aged mice

We first characterized the cardiac phenotype in a natural aging mouse model. Compared to young (3-month-old), aged mice (20-month-old) exhibited increased heart size (Fig. 1A), as demonstrated by higher heart weight/body weight ratio (HW/BW) (Fig. 1B) and heart weight/tibial length ratio (HW/TL) (Fig. 1C).

Figure 1 Cardiac phenotype in aged mice.

(A) Representative diagram of a mouse heart (left: 3 m, right:20 m). (B) Heart weight/body weight ratio (HW/BW) of mouse hearts. (C) Heart weight/tibia length ratio (HW/TL) of mouse hearts. (D–K) Cardiac ultrasound measurements of mouse hearts. (L) Representative diagrams of cardiac ultrasound of mouse hearts (top: 3 m, bottom: 20 m). Each dot represents an individual animal (n = 5 per group). Statistical significance was determined using an unpaired two-tailed Student’s t-test. All data are presented as mean ± standard error (SE). *P < 0.05, **P < 0.01. p-values are unadjusted.

Cardiac ultrasound examination was then performed on mice from both age groups. At comparable heart rates (Fig. 1D), aged mice showed no significantly differences in ejection fraction (EF) (Fig. 1E) or FS (Fig. 1F) compared to young mice. However, the Tei index, an indicator of global myocardial performance, was significantly elevated in the aged group (P < 0.01, Fig. 1G), indicating impaired myocardial function. Specific echocardiographic measurements revealed that, the IVSd (Fig. 1H) and LVPWd (Fig. 1K) were increased in aged mice. Further both left ventricle mass (Fig. 1L) and corrected heart mass (Fig. 1J) were significantly greater in the 20-month-old group. Other echocardiographic parameters showed no significant difference between groups (extended Fig. 1).

Histologic alterations in the heart of aged mice

Histological examination was performed on left ventricles sections from both 3-month-old and 20-month-old mice. HE staining revealed that the myocardium of young mice exhibited neatly arranged myocardial fibers, uniformly distributed nuclei, and intact cellular morphology (Fig. 2A). In contrast, myocardium from aged mice showed disrupted alignment of myocardial fibers, enlarged cardiomyocyte volume, and irregular cellular arrangement, indicating significant structural remodeling (Fig. 2B).

Figure 2 Histologic alterations in the heart of aged mice.

(A–B) HE staining of cardiac tissue structures. Representative histopathologic analysis of left ventricular heart sections stained with HE. Hematoxylin stains basophilic structures to a blue-purple color, and eosin stains acidophilic structures to a bright pink color. (A) 3-month-old mouse heart sections. (B) 20-month-old mouse heart sections (scale bar = 20 µm). (C–D) Masson staining of cardiac tissue structures. Representative histopathologic analysis of left ventricular heart sections stained with Masson. Myocardial tissue was stained pinkish-purple, and collagen fibers were stained blue. (C) Heart sections of 3-month-old mice. (D) 20-month-old mouse heart sections (scale bar = 20 µm). (E) Collagen volume occupancy plot (N = 3). (F) TUNEL staining of cardiac tissue structures. Representative micrographs of TUNEL-stained cells of myocardial tissue (FITC), corresponding total cell micrographs (DAPI), and their merged micrographs (Merge). TUNEL staining of representative apoptotic cells in green; nuclei stained by DAPI in blue (scale bar = 20 µm). (G) Percentage of TUNEL-positive cells (N = 3). (H–I) β-galactosidase staining of cardiac tissue structures. (H) Representative photographs of β-galactosidase staining of myocardial tissues from 3-month-old mice. (I) Representative photographs of β-galactosidase staining of myocardial tissues from 20-month-old mice (left: scale bar = 1,000 µm; right: scale bar = 20 µm). β-galactosidase positively stained cells were stained brown. (J) β-galactosidase staining density ratio (N = 3). Each dot represents an individual animal (n = 3 per group). Statistical significance was determined using an unpaired two-tailed Student’s t-test. All data are presented as mean ± SE. **P < 0.01 vs 3m. p-values are unadjusted.

Masson’s trichrome staining demonstrated increased collagen deposition within the myocardial tissue of 20-month-old mice compared to young controls, reflecting enhanced myocardial fibrosis during aging (Figs. 2C–2E). Additionally, TUNEL staining identified significantly increased cardiomyocyte apoptosis in the hearts of aged mice relative to young mice (Figs. 2F–2G). Furthermore, aged mice exhibited significantly senescence-associated β-galactosidase activity, indicating enhanced cellular senescence within the aged myocardium (Figs. 2H–2J).

Identification of candidate lncRNAs and mRNAs associated with cardiac aging

To systematically identify transcriptomic changes associated with cardia aging, we performed microarray-based transcriptome profiling of cardiac tissues from 3-month-old (young) and 20-month-old (aged) C57BL/6 mice. A total of 67,236 lncRNA probes and 16,720 mRNA probes were detected. Differential expression analysis revealed 1,293 upregulated and 1,551 downregulated lncRNAs in aged hearts compared to young controls, as visualized by the volcano plot (Fig. 3A). Similarly, 663 mRNAs were upregulated and 1,597 were downregulated in aged hearts compared to young hearts (Fig. 3B). The top 100 differentially expressed lncRNAs and mRNAs are listed in extended Tables S1 and S2, respectively.

Figure 3 Identification of candidate lncRNAs and mRNAs associated with heart aging in mice.

(A–B) RNA sequencing analysis of aged and young mouse hearts. Volcano plots of lncRNAs (A) and mRNAs (B) were analyzed in aged and young mouse hearts. Red indicates up-regulated mRNAs or lncRNAs, green indicates down-regulated mRNAs or lncRNAs, and black indicates unregulated mRNAs or lncRNAs. (C–H) Enrichment analysis of differentially expressed mRNAs in aged and young mouse hearts. (I–J) KEGG pathway analysis of differentially expressed mRNAs in aged and young mouse hearts.

To further explore the biological significance of these transcriptional changes, we performed GO and KEGG enrichment analyses on the differentially expressed mRNAs. GO analysis indicated the upregulated mRNAs were significantly enriched in biological processes such as defense response, stress response, leukocyte cell–cell adhesion, inflammatory response, and negative regulation of endopeptidase activity (Fig. 3C). Regarding cellular components, upregulated mRNA was primarily associated with membranes, extracellular matrix, and cell surface structures (Fig. 3D). Enriched molecular functions included passive transmembrane transporter activity, channel activity, endopeptidase regulatory activity, and cyclin-dependent protease activity (Fig. 3E).

In contrast, downregulated mRNAs were predominantly enriched in biological processes related to cellular metabolic processes, cellular organization, and general cellular function (Fig. 3F). Cellular component enrichment highlighted structures such as organelles, cytoplasm, and intercellular junctions (Fig. 3G), whereas molecular function enrichment included catalytic activity and structural molecule activity (Fig. 3H).

KEGG pathway analysis further revealed that upregulated mRNAs were significantly involved in the inflammation- and stress-related signaling pathways, including NFκB signaling, IL-17 signaling, Hippo signaling, cytokine receptor interactions, and C-type lectin receptor signaling pathways (Fig. 3I). Conversely, pathways associated with the downregulated mRNAs in aged hearts encompassed key aging-related pathways such as mitogen-activated protein kinase (MAPK) signaling, ubiquitin-mediated proteolysis, ErbB signaling, insulin signaling, and autophagy (Fig. 3J).

PPI network reveals sparse connectivity among aging associated DEGs

Protein–protein interaction (PPI) analysis of the top 100 DEGs mRNA from aged mouse hearts revealed a sparsely connected network. Using the STRING database with a medium confidence threshold (score >0.4), we identified 100 nodes and only 29 edges, resulting in an average node degree of 0.58 (Fig. 4). This indicates that most genes have few or no direct interactions with each other, and that the DEGs do not form a cohesive or densely interconnected network.

Figure 4 Protein–protein interaction network of the top 100 DEGs generated using STRING.

The protein–protein interaction (PPI) network of the top 100 differentially expressed genes (DEGs) in aged mouse hearts constructed using the STRING database (confidence score >0.4). The network contains 100 nodes and 29 edges, with an average node degree of 0.58. Most nodes are sparsely connected, and many remain isolated, indicating low functional connectivity among the DEGs. Node size and color are proportional to the number of interactions (degree). No significant pathway enrichment was detected.

Furthermore, no statistically significant enrichment was detected for GO terms or KEGG pathways, suggesting that these DEGs are not confined to a specific biological process or molecular pathway.

This lack of connectivity reflects the functionally heterogeneous nature of gene expression changes during cardiac aging. Rather than being organized into a central signaling hub, the transcriptomic alterations appear to be distributed across a diverse range of biological processes. Consistent with this, our functional annotation of the top 20 DEGs (Extended Table S3) revealed involvement in immune regulation (Cxcl13, Il3ra, Chil3), mitochondrial function (Timm50), transcriptional control (Vgll2, Zfp65), cell adhesion (Iglon5), protease activity (Klk1b16, Wfdc21), and cell cycle regulation (Cenpu).

Notably, several of these gene are closely related to inflammaging, a hallmark of aging characterized by chronic low-grade inflammation (Ramos et al., 2025). Specifically, Cxcl13 (a chemokine mediating B-cell chemoattraction), Skint7 (T-cell selection), Tspan32 (immune cell surface signaling), Wfdc21 (a WAP-domain protease inhibitor), and Mmd2 (monocyte-to-macrophage differentiation) were identified as inflammaging-associated DEGs. These findings underscore the contribution of immune remodeling and persistent inflammation to the transcriptome landscape of the aging heart.

Discussion

Aging is a complex biological process influenced by multiple factors, including genetic predisposition, environmental exposures, and lifestyle behaviors. It is characterized by a gradual decline in physiological and functional capacities, leading to increased susceptibility to various diseases and ultimately death. Cardiovascular disease (CVD) is one of the most prominent aging-related conditions, with its incidence rising steadily with advancing age, regardless of gender. Recent data indicate that the overall crude incidence of CVD in 2020 was 29.01% higher compared to 2010 (Tsao et al., 2023). Given the rapid global increase in the elderly population, understanding, and addressing cardiac aging represents a critical challenge for contemporary healthcare systems worldwide.

Our study demonstrated structural and functional characteristics of cardiac aging in mice, consistent with previous finding (Lakatta & Levy, 2003; Spencer et al., 2004). Structurally, aged hearts exhibited higher HW/BW and HW/TL ratios, thickened IVSd and LVPWd dimensions. Functionally, although conventional indices of resting systolic function, such as EF and FS, were preserved, we observed a significantly elevated Tei index in aged hearts (Fig. 1). This finding suggests the presence of subclinical cardiac dysfunction, aligning closely with clinical observations in older adults who typically exhibit subtle systolic and diastolic impairments even prior to overt EF decline (Vasan et al., 2018). Furthermore, aging-related reductions in cardiac functional reserve capacity—a hallmark of cardiac aging—reflect the heart’s increased vulnerability to stress (Chiao & Rabinovitch, 2015), contributing to reduced exercise tolerance observed in elderly populations (Pandey et al., 2020). Such physiological changes are particularly relevant to heart failure with preserved ejection fraction (HFpEF), a predominant cardiovascular condition in the elderly individuals (Upadhya et al., 2017). Histological analysis further confirmed significant myocardial remodeling, including increased cardiomyocyte hypertrophy, apoptosis, fibrosis, and enhanced senescence-associated β-galactosidase activity (Fig. 2). Together, these structural and functional alterations highlight critical early markers and potential intervention targets in age-related cardiac dysfunction.

Next, our transcriptomic analysis identified numerous differentially expressed lncRNAs and mRNAs in aged hearts, revealing distinct biological functions and pathway implicated in cardiac aging. Interestingly, GO and KEGG analyses revealed distinct biological functions and signaling pathways associated with these transcriptomic shifts.

Upregulated mRNAs were significantly enriched for processes such as defense response, inflammatory response, leukocyte adhesion, and stress response (Fig. 3C), highlighting inflammation as a central theme. These findings align closely with the established concept of “inflammaging”, a chronic, low-grade inflammatory state accompanying biological aging (Wang & Shah, 2015; Baylis et al., 2013).

PPI network analysis further supported these findings, suggesting that cardiac aging involves a multidimensional and cell-type diverse transcriptomic shift, affecting numerous interconnected biological pathways rather than a single unifying mechanism (Fig. 4). Indeed, cardiac aging is increasingly recognized as a coordinated phenomenon involving multiple cell types. It is driven by complex crosstalk among metabolic, inflammatory, and epigenetic regulatory pathways (Chen, Lee & Garbern, 2022). Within this landscape, immune cells exert paradoxical effects: on one hand, macrophages and T cells help maintain cardiac homeostasis by removing senescent cardiomyocytes (Song, An & Zou, 2020); on the other hand, during chronic inflammatory conditions, immune cells secrete pro-inflammatory and pro-fibrotic factors (e.g., MMP9, TNF-α), exacerbating cardiomyocyte hypertrophy and fibrosis, thus accelerating cardiac aging (Horckmans et al., 2017).

Macrophages, in particular, have been highlighted as critical mediators of cardiac inflammaging. Age-associated alterations in macrophage function, including impaired phagocytic activity (Khare, Sodhi & Singh, 1996) and elevated secretion of immunosuppressive prostaglandin E2 (Beharka et al., 1997), collectively contribute to immune dysfunction and increase vulnerability to cardiovascular diseases (Mahbub, Deburghgraeve & Kovacs, 2012). Hulsmans et al. reported increased cardiac macrophage infiltration in both humans and mice with diastolic dysfunction, a common condition associated with aging and hypertension. In such states, macrophages secrete interleukin-10 (IL-10), activating fibroblasts, promoting collagen deposition, and consequently impairing myocardial relaxation and increasing myocardial stiffness. Importantly, targeted deletion of macrophage-derived IL-10 has been shown to ameliorate diastolic dysfunction (Hulsmans et al., 2018). These observations underscore cardiac macrophage expansion and their phenotypic alterations as potential therapeutic targets for mitigating cardiac fibrosis and associated diastolic dysfunction in aging hearts.

In contrast, down-regulated mRNAs revealed in aged hearts were involved primarily in fundamental cellular functions, such as metabolism, cellular organization, and maintenance of structural integrity (Fig. 3F). These changes reflect a generalized impairment in cellular and metabolic functions with aging, collectively impairing cardiomyocyte viability and overall cardiac performance.

LncRNAs, extensively identified in our analysis, represent another important layer in age-related cardiac remodeling. Recent studies have emphasized lncRNAs as critical regulators of inflammaging and potential therapeutic targets in cardiac aging. Mechanistically, lncRNAs influence the aging heart by regulating complex miRNA networks, inflammatory pathways such as NF-κB and STAT3 signaling, and cellular senescence genes like p53 and p21, as systematically summarized previously (e.g., H19, Meg3, MALAT1) (Varghese, Schwenke & Katare, 2023). However, despite these insights, many functions, and regulatory mechanisms of lncRNAs remain poorly defined. The candidate lncRNAs identified in our study thus represent valuable targets for future mechanistic exploration. Collectively, further characterization of these lncRNAs may uncover novel therapeutic strategies to mitigate cardiac aging and enhance cardiovascular health in aging populations.

Our results significantly expand current knowledge by identifying novel candidate lncRNAs and mRNAs potentially involved in cardiac aging. Nevertheless, this study has several limitations.

First, the descriptive nature of our analysis necessitates further mechanistic validation through functional experiments to confirm the roles of identified genes in age-associated cardiac dysfunction.

Second, the study was conducted exclusively on male mice to minimize hormonal variability. However, this limits the generalizability of our finding, especially considering increasing recognition of sex-based differences in aging and cardiovascular biology. Future studies incorporating both sexes are crucial to validate these observations and to uncover potential sex-specific regulatory mechanisms.

Third, we did not perform independent experimental validation using RT-qPCR or protein-level assays. While microarray analysis provides broad transcriptome coverage and robust comparative expression data, it lacks the quantitative precision of orthogonal methods. Additional validation using RT-qPCR or Western blotting would improve the reliability of our results.

Fourth, microarray technology itself has inherent limitations, including its inability to detect novel transcripts, limited sensitivity to low-abundance RNAs, and reliance on pre-designed probe sets. These constraints may prevent the detection of unannotated or low-expressed regulatory lncRNAs relevant to cardiac aging.

Fifth, the current analysis remains descriptive in nature and does not include network-level bioinformatics approaches such as competing endogenous RNA (ceRNA) network construction or weighted gene co-expression network analysis (WGCNA). Such analyses could reveal regulatory modules and key molecular hubs, particularly for lncRNAs implicated in inflammaging. The candidate lncRNAs identified in our dataset therefore represent promising targets for future mechanistic and network-level studies.

Lastly, a longitudinal, time-course study incorporating intermediate stages of aging could provide more dynamic insights into transcriptomic shifts across the aging continuum. Although these approaches were beyond the scope of the current study due to resource constraints, future research integrating temporal sampling, sex-based comparisons, network-level modeling, and experimental validation will be critical for delineating the molecular architecture of cardiac aging and for identifying therapeutic targets.

Despite the limitations, this study possesses several notable strengths. First, we employed a well-characterized natural aging mouse model, which closely reflects physiological cardiac aging in humans, avoiding the confounding effects of disease- or stress-induced models. We integrated multilevel phenotyping, including echocardiographic assessment, histological staining, and transcriptome profiling, to comprehensively capture structural, functional, and molecular alterations associated with cardiac aging.

Second, to our knowledge, this is among the few studies that simultaneously analyzed lncRNA and mRNA expression in naturally aging hearts using a lncRNA-specific microarray platform. Our systematic annotation of differentially expressed genes—including enrichment of inflammation-related pathways and immune modulators—provides novel insights into the inflammaging hypothesis and supports the emerging role of immune–epigenetic interactions in cardiac aging.

Third, we publicly deposited the full dataset in the Gene Expression Omnibus (GSE256034), and applied validated analytical pipelines using limma and clusterProfiler, ensuring transparency, reproducibility, and broad utility for future hypothesis generation and validation efforts.

Taken together, these features highlight the originality, technical rigor, and translational relevance of our work, while laying a valuable foundation for future studies aimed at understanding and intervening in the molecular mechanisms of cardiac aging.

Conclusion

Age-associated lncRNAs and mRNAs were identified by studying naturally aging mice’s cardiac phenotype. Potential molecular-based diagnostic and therapeutic applications of these lncRNAs and mRNAs could be explored.

Protocol Registration

No formal protocol registration was performed prior to the start of this study. However, a detailed internal study plan outlining the research question, experimental design, and analysis strategy was prepared in advance and approved by the Animal Ethics Committee of Zhongshan Hospital, Fudan University (Approval No. 2021-20). As this was an exploratory, non-interventional study involving comparison of age-related differences in mice, and protocol registration is not mandated for such preclinical studies, no external registration was conducted.

Supplemental Information

Supplemental Information 1 Cardiac ultrasound measurements of mouse hearts

Supplemental Information 2 Top 100 differentially expressed lncRNAs in mice hearts (20 m vs. 3 m)

Supplemental Information 3 Top 100 differentially expressed mRNAs in mice hearts (20 m vs. 3 m)

Supplemental Information 4 Top 20 DEG mRNAs Functional Annotation

Supplemental Information 5 ARRIVE 2.0 checklist

Supplemental Information 6 Raw data

Abbreviation

lncRNA long non-coding RNAs

HE hematoxylin-eosin

IVSs interventricular septum in systole

LVPWd left ventricular posterior wall in diastole

ncRNAs non-coding RNAs

miRNAs microRNAs

SPF specific pathogen-free

HR heart rate

IVSd interventricular septum in diastole

LVIDs left ventricular internal diimension in systole

LVIDd left ventricular internal diimension in diastole

LVPWs left ventricular posterior wall in systole

LVEF rejection fraction

FS fractional shortening

MPI or Tei index myocardial performance index

IVRT isovolumic relaxation time

IVCT isovolumetric contraction time

ET ejection time

PFA paraformaldehyde

TUNEL terminal deoxynucleotide transferase dUTP nick end labeling

FPKM fragments per kilobase of exon model per million

DEGs differential expression genes

GO Gene ontology

KEGG Kyoto Encyclopedia of Genes and Genomes

PPI protein–protein interaction

HW/BW heart weight/body weight ratio

HW/TL heart weight/tibial length ratio

CVD cardiovascular disease

HFpEF heart failure with preserved ejection fraction

IL-10 interleukin-10

ceRNA competing endogenous RNA

WGCNA weighted gene co-expression network analysis

Additional Information and Declarations

Competing Interests

Author Contributions

Animal Ethics

Microarray Data Deposition

Data Availability

The authors declare there are no competing interests.

Zheng Kuai conceived and designed the experiments, prepared figures and/or tables, authored or reviewed drafts of the article, and approved the final draft.

Zheng Li performed the experiments, prepared figures and/or tables, and approved the final draft.

Jianguo Jia performed the experiments, prepared figures and/or tables, and approved the final draft.

Yongle Chen performed the experiments, prepared figures and/or tables, and approved the final draft.

Xiaoyi Zhang analyzed the data, prepared figures and/or tables, and approved the final draft.

Jianhui Zhang analyzed the data, prepared figures and/or tables, and approved the final draft.

Yangli Ye conceived and designed the experiments, prepared figures and/or tables, and approved the final draft.

Lihong Gao conceived and designed the experiments, prepared figures and/or tables, and approved the final draft.

Ling Li conceived and designed the experiments, prepared figures and/or tables, and approved the final draft.

Yu Hu performed the experiments, authored or reviewed drafts of the article, and approved the final draft.

The following information was supplied relating to ethical approvals (i.e., approving body and any reference numbers):

The animal research protocol approved by the Animal Ethics Committee of Zhongshan Hospital, Fudan University (Approval No. 2021-20).

The following information was supplied regarding the deposition of microarray data:

The microarray datasets are available at GEO: GSE256034.

The following information was supplied regarding data availability:

The microarray datasets are available at GEO: GSE256034.

All analysis scripts used to generate the figures and Supplemental Tables are available at Figshare: Kuai, Rachel (2025). Code. figshare. Dataset. https://doi.org/10.6084/m9.figshare.30294484.v1.

The raw data is available in the Supplemental File.

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
