# Peer review of "Identification of candidate long non-coding RNAs and mRNAs associated with heart aging in mice"

_PeerJ, doi:10.7717/peerj.20433_

## Round 0.1 · original submission · Major Revisions

· Academic Editor

Major Revisions

The volcano plots are too low resolution so please ensure you provide high-quality images with your revision.

**Language Note:** The review process has identified that the English language must be improved. PeerJ can provide language editing services - please contact us at [email protected] for pricing (be sure to provide your manuscript number and title). Alternatively, you should make your own arrangements to improve the language quality and provide details in your response letter. – PeerJ Staff

Reviewer 1 ·

Basic reporting

This is an interesting study of mouse heart ageing looking at physiological as well as transcriptomic differences between 3 and 20 month old mice. The writing is clear and straightforward with few English corrections. "Notably" occurs twice on line 311 is the only typo I saw. The main problem I have with this manuscript is they mention "RNA sequencing"/high throughput sequencing while it appears they use microarrays for transcriptome quantitation. I'm still not sure if they used RNA sequencing as well as microarrays? This needs to be corrected and clarified. Which protocol was used, or were both? Similarly line 202, "their predicted targets", I'm not sure what the targets are? Drug targets of protein coding genes? Also they state the mice were house individually on line 82 but then say cage density of 3-4 mice on line 87. These sort of obvious typos are more likely when AI writing has been used to generate the text.

At a scientific level it would be nice to see a bit more analysis of the lncRNAs, but this may be beyond the scope of their project. For instance co-expression analysis of lncRNAs might give some more information on their function. Even looking at the top 10 lncRNA functions I can see a host of interesting aspects to write about that would flesh out what the differential transcripts are doing/reacting to and feeding nicely into the inflammaging hypothesis they talk about. The authors may want to add an expert in cellular biology to pad out the discussion in the paper, or at least google the top 10 or so lncRNAs and discern their function in relation to the inflammaging hypothesis. There is some good results for them there!

Experimental design

Experimental design is sufficient for a preliminary investigation of lncRNAs in mouse tissue.

Validity of the findings

The data the authors generate is interesting and useful for the scientific community. The transcriptome quantitation method needs clarification. Their results are intriguing and it would be a pity not to discuss the differential genes expressed in more detail (GO only GOes so far..), or even expand the analysis with the help of a bioinformatician. string-db.org and some digging round into the function of the top 10 genes would add a lot.

Reviewer 2 ·

Basic reporting

In this manuscript, Kuai and colleagues performed lncRNA and mRNA profiling using RNA microarrays to decipher changes in aging mice. They compared wild-type young (3-month-old) and aged (20-month-old) mice by utilizing echocardiography, histology, cellular-based assays, and transcriptome-level analysis.
Considering the importance of aging in the development of cardiovascular diseases, this is an interesting topic with significant potential. However, I strongly believe that the study requires a more in-depth analysis and additional effort before it is ready for publication.
I understand that the authors are not native English speakers; however, the overall language throughout the manuscript should be revised to make the study clearer and more comprehensible. Some sections, as outlined below, need particular attention and rephrasing to avoid over-interpretation of the results.

Experimental design

The overall structure of the manuscript is clear; however, some fundamental revisions are required.
To begin with, the authors should state the aim of the study more clearly in the “Introduction” section. They aimed to investigate structural and transcriptional changes in the aging mouse heart, but it is not clear why they chose to analyse lncRNAs and mRNAs separately from the same dataset. Is there a specific rationale for this approach? Considering that lncRNAs and mRNAs share similar characteristics, the authors should provide more detail on the analysis pipeline, including which software and packages were used to perform the sequencing analysis. I thank authors to provide supplementary tables with the processed RNA data, however, how the analysis is performed is completely missing should be detailed.

A similar suggestion applies to the entire “Methods” section: wherever methods are described, the chemicals or kits used should be stated clearly, including catalogue numbers and company names. The instrument used for echocardiographic data acquisition, as well as the software employed for analysis, should also be clearly stated.

Validity of the findings

This study has great potential, and I offer the following suggestions to improve the quality of the work:
• The study includes only male mice; including females would provide more robust data and offer an opportunity to investigate sex differences in aging.
• Additionally, a time-course RNA sequencing would also increase the value of the study (3 months – 20 months and additional age in between). Even to confirm their findings in RT-qPCR to observe the trend would be helpful.
• For cardiac ultrasound, the use of measurement parameters should be consistent. For example, if reporting interventricular septal thickness in systole or diastole (IVSs or IVSd), the terminology should remain consistent throughout the study. The same applies to LVPWs or LVPWd.
• There are many similar studies with similar datasets, authors should consider to integrate/ compare their findings with those publicly available data sets to show their study has a value for confirmation of the results, or identify additional RNAs might have role.

Additional comments

Figure 1:
• Please indicate each animal as a single dot in the bar plots.
• The figure legend should include more detail, particularly specifying which statistical tests were used.
Figure 2:
• The images should be labelled more clearly.
• A scale bar should be added to each image.
• Each animal should be represented individually in the plots.
Figure 3:
• The figure is not readable—I was unable to interpret any of the plots. The image quality needs to be significantly improved.

Minor Comments:
Line 258: Please revise the phrase “to determine the mechanistic role…” — the study is primarily descriptive/profiling in nature and does not directly investigate mechanistic pathways.
Line 24: There is a duplication of the word “novel” in the first sentence of the abstract — this should be corrected.
Section 2.1 and 2.2: This section is somewhat repetitive and could be summarized more concisely without losing essential information.

·

Basic reporting

The Manuscript by Kuai Z et al describes intersting study in which the Authors compared younger with older mice. The methodology was applied correctly (histological, echocardiographic study and RNA microarray study) and the results were conclusive. Nevertherless, the main shortcoming of the Manuscript is the background - the Authors should precisely indicate what's new in their paper, They should provide information (supported with appropriate references) what is already known regading aging in mice, are there any similiar studies on mice aging published before (e.g. histological or echocardiographical) - I mean, studies which compared younger with older mice.
I'd suggest to provide information about metaanalyses that had already been done.

Experimental design

In the experimental design it should be indicated what was the time interval between echocardiography (with anesthesia by isoflurane) and mice euthanasia.

Validity of the findings

It should be emphasized that in this study only microarrays were undertaken, there were no more detailed study (e.g. by qRT-PCR) targeted at specific mRNAs or lncRNAs undertaken on larger group of animals. Therefore, the conclusions should be made with caution and, as I mentioned in "Basic reporting", the Authors should emphasize the novel aspects of their study.

Additional comments

There are some linguistic and typing errors, e.g. duplications of words (like "novel novel") in the abstract or "notably" in Discussion (page 21).

---

## Round 0.2 · Minor Revisions

· Academic Editor

Minor Revisions

Thank you for addressing the prior reviewer comments. Before moving to a decision, the following items need to be addressed: 1. What multiple hypothesis test correction was used? Please clearly label all reported p-values with the test for generating those p-values and the correction method. 2. Data and code must be available. Thank you for making the data available on GEO, but I do not see a link to the code to replicate the figures and findings (e.g., a DOI through Zenodo or FigShare).

---

## Round 0.3 · accepted · Accept

· Academic Editor

Accept

I have assessed the manuscript myself. Thank you for addressing the prior concerns. I believe the manuscript is now acceptable for publication; however, you should add the FigShare code link to the final manuscript.